# Anti-SARS-CoV-2 Vaccination Campaign: Risk Perception, Emotional States, and Vaccine Hesitancy in a Sample of Adolescents’ Vaccinated Parents in Southern Italy

**DOI:** 10.3390/vaccines10060958

**Published:** 2022-06-16

**Authors:** Giulia Savarese, Luna Carpinelli, Anna De Chiara, Claudio Giordano, Matilde Perillo, Domenico Fornino, Francesco De Caro, Mario Capunzo, Giuseppina Moccia

**Affiliations:** Department of Medicine and Surgery, University of Salerno, Baronissi Campus, 84081 Baronissi, Italy; gsavarese@unisa.it (G.S.); lcarpinelli@unisa.it (L.C.); anna_de_chiara@libero.it (A.D.C.); clgiordano@unisa.it (C.G.); maperillo@unisa.it (M.P.); dfornino@unisa.it (D.F.); mcapunzo@unisa.it (M.C.); gmoccia@unisa.it (G.M.)

**Keywords:** anti-SARS-CoV-2 vaccination, risk perception, emotional states, vaccine hesitancy, adolescents’ parents

## Abstract

Background: The international strategic plan for COVID-19 vaccines remains the practical option for the protection of health. However, vaccine hesitancy remains an obstacle to full population vaccination, with rapid developments in COVID-19 vaccines and concerns about efficacy acting as influencing factors. Aim: The present study investigated the perception of vaccine hesitancy among parents of adolescents in order to explore the reasons and related emotional states. Methods: In January–March 2022, an online questionnaire was administered to a sample of parents who brought their children to the vaccine center of a local health unit, ASL Salerno (Campania, Italy). Results: The participants were 1105 parents (F = 64.6%; mean age = 47.37 years, SD = 7.52) of adolescents (F = 47.6%; mean age = 14.83 years, SD = 1.72). All parents had received the COVID-19 vaccine. Regarding the vaccination schedule, 46.8% believed that children receive more vaccinations than they should; 25.1% believed that it is better to develop immunity rather than get vaccinated; 41.2% believed that their child could have side effects; 29.6% were very concerned that vaccines were unsafe, while 35.3% believed vaccines do not prevent disease; 21.5% were very reluctant about pediatric vaccines; and 23.8% did not trust the information received. Conclusions: In order to increase vaccination and reduce the prevalence of vaccine hesitancy, it is essential to support the value of vaccination among all parents and make information more accessible and usable through competent pediatricians.

## 1. Introduction

The term “vaccine hesitancy” has been defined by the World Health Organization (WHO) Strategic Group of Experts for Vaccination (SAGE) as a delay in accepting or refusal to accept the offer of vaccination, despite the availability of services [1,2,3]. The WHO study group, established in 2012 [4], indicates that vaccine hesitancy is a phenomenon that is not limited geographically or to specific contexts; worldwide, there is a worrying increase in critical attitudes toward vaccination, which was once received as a sign of progress and a healthcare right. Along with being extensive, the phenomenon is also complex and characterized by various factors in different contexts (political, ideological, social, etc.) [5]. In general, vaccination hesitancy describes people’s reluctance to accept the vaccination offer. It is clear that the problem of vaccination hesitancy was considered relevant long before the advent of COVID-19. In fact, the WHO, before declaring the pandemic, listed it among the 10 main threats to global health [6]. A few months after the spread of the SARS-CoV-2 virus in the West, studies on the subject followed one after another with some regularity, reporting fluctuating percentages of vaccination hesitancy among various countries. To get an updated picture of the situation as much as possible, we can refer to a recent and concise systematic review that appeared in the journal *Vaccines* [7]. The acceptance rates of COVID-19 vaccines among 33 countries were particularly high in Ecuador (97.0%), Malaysia (94.3%), Indonesia (93.3%), and China (91.3%), and particularly low in Kuwait (23.6%), Jordan (28.4%), Italy (53.7%), Russia (54.9%), Poland (56.3%), the United States (56.9%), and France (58.9%). In addition, high vaccination hesitancy rates were found among students and health care professionals [8].

The approval of vaccination for adolescents requires greater attention, in order to achieve high immunization rates within adolescent populations. Immunization within these populations, in fact, has the advantages of both protecting children and adolescents from morbidity and mortality, and reducing the spread of the virus among clinically vulnerable people. The epidemiological report of the Italian Higher Institute of Health [9] released on 26 November 2021 highlighted an increase in the incidence in the entire age group 0–19 years, but in particular, in the population under the age of 12, which was, at that time, still not eligible for vaccination and showed a higher incidence than other age groups. Since the beginning of the epidemic, in the population 0–19 years, there have been 826,774 confirmed cases of infection, with 8632 hospital admissions, 251 visits to intensive care, and 35 deaths.

On 24 November 2021, the Committee for Medicinal Products for Human Use (CHMP) of the European Medicines Agency’s (EMA) recommended granting an extension of indication for Comirnaty, the COVID-19 vaccine, to approve its use in children aged 5 to 11 years. The vaccine, developed by BioNTech and Pfizer, is already approved for adults and children from 12 years of age. In the age group 5–11 years, the dose of Comirnaty will be lower than that used in people aged 12 years and older (10 µg compared to 30 µg). As in the case of the age group ≥12 years, the vaccine is administered as two injections into the muscle of the upper arm, three weeks apart. A study in children aged 5 to 11 years showed that the immune response to Comirnaty, given at a lower dose (10 µg) in this age group, was comparable to that seen at the higher dose (30 µg) in the age group between 16 and 25 years (measured by the level of antibodies against SARS-CoV-2). This recommendation was also confirmed by the Italian Medicines Agency (AIFA) on 1 December 2021.

Research studies aimed at investigating COVID vaccination hesitancy among adults have highlighted some factors related to concern about the side effects and adverse events associated with vaccines, their rapid authorization, and mistrust of governments and health institutions [10,11]. Available data on parental vaccine hesitation versus COVID vaccination was collected when vaccines were not authorized for use in pediatric subjects [12,13].

Although studies have shown that vaccines against COVID-19 are safe and effective, some studies have shown vaccination hesitancy among children and adolescents and their families [14].

Based on a survey conducted between 15 April and 23 April 2021 in the United States, only 55.5% of 1022 parents and guardians of unvaccinated teens aged 12–17 reported they would “definitely” or “probably” allow their child to receive a COVID-19 vaccine, and only 51.7% of 985 adolescents aged 13–17 would “definitely” or “probably” receive a vaccine [15]. Vaccination hesitancy was found to be more frequent in families with individuals who have allergies, including those with asthma, who were therefore fearful of an allergic reaction to the vaccine. A systematic review and meta-analysis found that the incidence of an allergic reaction to a COVID-19 mRNA vaccine is 7.9 cases per million doses (95% CI 4.02–15.59). In fact, the studies conducted so far show that allergic reactions resolve quickly without long-term sequelae [16]. The survey also found that vaccination hesitancy was more common among respondents whose level of education was less than a university degree.

Hesitancy to vaccinate against COVID-19 has been found worldwide by several studies [17,18]. Specifically, in Italy, some studies have been conducted on vaccination hesitancy among parents [19], and it was found that 12.4% were highly hesitant toward anti-COVID-19 vaccination, and that some variables such as level of schooling, usefulness of the vaccine, and quality and quantity of information received are related to a higher likelihood of being highly hesitant. In addition, an analysis was offered by Gallé et al. [20] on COVID-19-influenced attitudes and quality of life; the authors found that although participants showed a good level of knowledge about COVID-19 and its prevention, they reported an increase in unhealthy habits that may have had important consequences for long-term health. To preserve the effectiveness of national vaccination campaigns, it is necessary to consider and address aspects of the general population’s attitudes and beliefs about vaccination and to adopt strategies that leverage behavioral and communication sciences. Therefore, the purpose of this study was to survey the variables and factors that underlie vaccination hesitancy among parents of adolescents in an as-yet unexplored territorial context, in order to highlight indications that might prove effective in developing an awareness of the issues that constitute the strengths of a vaccination campaign, and in developing targeted interventions according to behavioral science techniques and appropriate communication principles and tools.

## 2. Methods

### 2.1. Setting Procedure

The present study involved parents who accompanied their adolescents to receive a COVID-19 vaccination at the Vaccination Center of the University Hospital ASL Salerno (Italy). During both the waiting period for inoculation and the post-vaccine observation phase, the participants were asked to fill in a questionnaire by scanning a QR code that directed them to a link on the Google Forms platform. Preliminarily, each participant was informed of the research purpose and, through informed consent, the use of data in anonymous and aggregate form. Each parent who participated in the survey accompanied a single vaccination recipient. The data were collected during January–March 2022.

### 2.2. Tools

The survey was conducted through the use of a questionnaire created ad hoc and placed on the Google Forms platform. The questionnaire was made up of four sections:Sociodemographic information: The subject was asked for variables such as sex, age, education, type of work, degree of kinship with the adolescent being vaccinated, and information relating to their own health in relation to COVID-19.COVID-19 experience: The subject was asked questions relating to their emotional states, thoughts, and behaviors in relation to the COVID-19 pandemic.Child Health Questionnaire (CHQ-PF50) [21]: This is a standardized and validated questionnaire that allows parents to assess their child’s well-being and health and to self-assess how much the severity of the disorder affects their emotional state, their organization of time, and the family unit. The CHQ-PF50 shows internal consistency equal to a Cronbach’s alpha of 0.39–0.96 (mean 0.72).Parental Attitudes About Childhood Vaccines (PACV) [22]: This is a questionnaire that evaluates parents’ attitudes toward vaccinations. It contains 15 multiple-choice items, and the validity and reliability of the instrument are good (Cronbach’s alpha = 0.70). The score ranges from 0 to 100 points, interpreted as follows: 0–29, low level of vaccine hesitancy; 30–49, intermediate level; and >50, high level.

The questionnaire was structured by expert psychologists, virologists, and epidemiologists through a preliminary analysis of the scientific literature, and dedicated focus groups structured the survey instrument and additional questions in Section 1 and Section 2.

### 2.3. Participants

The Vaccination Center of the University Hospital ASL Salerno covers, for health reasons, the city of Salerno (Campania, Italy), which has a territorial area of 59.85 km^2^ and a population of 127,765 inhabitants (data updated to March 2022). The sampling used for the present study is of “convenience”. A total of 1105 parents (64.6% F; mean age = 47.50 years, SD = 7.65) of adolescents waiting for a COVID-19 vaccine completed the questionnaire. The reference sample consisted of 64.5% mothers, 31.4% fathers, and 4.1% other family members (uncles, grandparents, brothers, or sisters). With regard to marital status, 75.3% of the parents were married, 8.4% separated, 7.9% single, 4.8% divorced, and 3.6% cohabiting. The average number of members per family unit was 3.94 (SD = 0.943). The level of schooling was as follows: 45.6%, secondary school diploma; 32.1%, university degree; 17.3%, lower secondary school diploma; and 5%, other qualifications. In relation to the type of work, 21% were employed by a public or private company, 20.8% were housewives, 16.3% were office workers, 15.8% were freelance professionals, 12% had other types of jobs, 7.9% were unemployed, and 6.2% were workers. At the time of the interview, the parents had already received the COVID-19 vaccine; specifically, 82.1% had a booster dose, 15% had completed the first vaccination course, and only 2.9% had only received the first dose. Among the sample, 13.6% had contracted the virus, 7.7% in asymptomatic form and 7.6% symptomatic. In reference to the vaccinated adolescents, 51.9% of the sample was male (mean age = 14.76 years, SD = 1.73) and 83.5% were getting a booster dose. All characteristics and descriptive variables of the sample are listed in Table 1.

### 2.4. Statistical Analysis

IBM SPSS Statistics 23.0 software was used for statistical analysis. An exploratory factor analysis was then conducted (EFA) to observe the factor structure, which was then confirmed through confirmatory factor analysis (CFA). The total of explained variance was 49.7%. The CFA values are greater than 0.90 and included in the confidence interval. Therefore, all indices considered confirm the results of the EFA. Data conforming to the normal distribution in descriptive statistics are presented as mean (M) ± standard deviation (SD). A comparison was made between the means of standardized scores obtained in the CHQ-50 and PACV tests. Cross-tabs were performed based on the most significant variables.

Differences in numerical data between sample groups were analyzed using ANOVA.

## 3. Results

### 3.1. COVID-19 Experience

With regard to the second section of the survey questionnaire, the average percentages of frequency of answers to questions related to the COVID-19 experience were determined to assess moods, concerns, and attitudes (*p* value < 0.001) (see Table 2). The analysis of the results showed that 36.8% of the parents were “quite worried” about problems related to the pandemic, while 17.7% were undecided; 38.3% were “quite worried” about being directly and personally affected by the pandemic in the next six months, while 20.2% were undecided; 40.1% were “quite worried” that their family and friends would be directly affected by the pandemic, while 20.1% were undecided; 16.9% said they “extremely disagree” with the claim that they were probably ill with COVID-19; only 18.2% said they “extremely disagree” that getting sick with COVID-19 is a serious problem; and 37.1% said they “extremely disagree” with the claim that COVID-19 will affect the population in Italy.

### 3.2. Child Health Perceptions

Generally, the evaluation of parents’ perceptions of the health of their adolescents awaiting vaccination showed the following (*p* value < 0.001): 44.8% said that their child’s health in the last 3 months was good, 33% said it was very good, 18.3% said it was excellent, 3% said it was passable, and 0.9% said it was poor.

Among the parents, 2.6% reported that there were health problems that severely limited the adolescents’ physical state, and 10.5% reported that emotional and behavioral problems greatly limited their health. Furthermore, 33.2% reported that behavioral problems such as arguments, aggression, and difficulty concentrating occurred “sometimes”; 10.2% reported that they often experienced sadness and a tendency to cry, while 16% reported nervousness and outbursts of anger; 5.5% reported never having felt happiness or satisfaction (4%); and 24% reported feeling a general satisfaction with life “sometimes”. Additionally, 8.7% reported daily anxiety and concern about their child’s physical health, and 7.1% about emotional health.

### 3.3. Vaccine Hesitancy

With regard to vaccine hesitancy (see Table 3), through the PACV scoring it was possible to determine the ranges of low (0–29), intermediate (30–49), and high (>50) levels in the general sample. Among the parents, 89.4% had a low level of vaccination hesitancy, 10.5% had an intermediate level, and 0.1% had a high level.

Specifically, a comparison (mean ± standard deviation; *p* value < 0.001) was made between the scores obtained on the PACV with reference to the categories/variables of belonging (sex, parenting, school level, type of work). For the gender variable, women had higher mean scores than men (23.88 ± 5.62 vs. 22.73 ± 4.39). With reference to the parenting variable, mothers had higher scores (23.88 ± 5.65) than fathers (22.60 ± 4.26). For the variable level of school education, participants with a lower middle school education had higher average scores (24.51 ± 4.54) than those who had a secondary school diploma (23.78 ± 6.51), other qualifications (23.20 ± 4.10), or a university degree (22.52 ± 4.10). For the employment variable, participants who were housewives (24.20 ± 4.52) or unemployed (24 ± 4.08) showed higher means than those who were workers (23.96 ± 3.91), employed by a public or private company (23.30 ± 4.66), office workers (22.90 ± 7.90), or freelance professionals (22.58 ± 4.53).

Furthermore, we analyzed some specific items of the PACV. From the descriptive analysis of the response frequencies for the general sample (*p* value < 0.05), it was found that 14.5% delayed vaccination, and 11.2% did not vaccinate their children for reasons other than allergies and diseases; 44.7% fully agreed that children are given more vaccines than needed for actual well-being, and 20.7% believed that vaccines do not prevent serious diseases; 22.9% strongly believed that it is more useful for their child to be immunized by contracting the disease than getting vaccinated; and 34.1% were very concerned that vaccines do not prevent disease. With regard to another child, 24.8% said they would not carry out the recommended pediatric vaccinations. Overall, 20.8% considered themselves highly hesitant about pediatric vaccinations, 21.6% said they did not trust the information on vaccines, and only 37.9% of the sample said they trusted their pediatrician.

## 4. Discussion

Our results highlight a very important aspect that, although present in the scientific literature, still reveals many gaps in terms of optimizing the decision-making process regarding vaccination to protect people according to a general ontogenetic orientation, which implies the idea of a psychological science of meaningful human conduct built by a goal-oriented agent [23]. From our sample, the results show that 8.7% suffered daily from anxiety and concern over the physical health of their children, and for 7.1%, this daily concern also included the emotional side. According to some data published in a report by the European Commission, “Europeans’ attitudes towards vaccination” [24], which investigated people’s attitudes about and knowledge of vaccines, only 85% of citizens in Europe believe that vaccines are effective in preventing infectious diseases (78% in Italy). This data is combined with the satisfaction related to the perception of the risk of the disease by a person. If there is a perception of low risk in terms of the threat posed by a vaccine-preventable disease, a person may show a limited desire and intention to vaccinate [25]. This antecedent, therefore, is linked to one specific disease, although individual factors such as age, health, and responsibilities can also affect levels of complacency. Complacency is also influenced by the individual’s perception of his or her own self-efficacy or one’s ability to do something to be vaccinated.

Almost half the population is afraid of serious side effects (48% in Europe vs. 46% in Italy). There is also little general awareness of the risks associated with vaccination-preventable diseases. Among the information obtained, the persistence of incorrect knowledge is apparent, a situation that leads to a loss of confidence in vaccinations. In this sense, it is worrying that one-third of Italians (32%) have the mistaken belief that vaccines weaken the immune system or can cause the disease they protect against (34%).

In fact, among our survey sample, 14.5% delayed vaccination and 11.2% did not vaccinate their children for reasons other than allergies and diseases; 44.7% fully agreed that children are given more vaccines than are needed for actual well-being; 20.7% believed vaccines do not prevent serious illness; 22.9% firmly believed that it is better for their child to get immunized by contracting the disease than getting vaccinated; and 34.1% were very concerned that vaccines will not prevent disease. With regard to another child, 24.8% said they would not obtain the recommended pediatric vaccinations. Overall, 20.8% considered themselves very reluctant with regard to pediatric vaccinations, 21.6% did not trust the information on vaccines, and only 37.9% trusted their pediatrician.

In a study on the relationship between vaccine hesitancy and health perception among mothers [26], the results showed that previous vaccine hesitancy attitudes and behaviors did not fully capture their acceptance of the COVID-19 vaccine or perception of the COVID-19 threat. Perception of the threat of COVID-19 influenced mothers’ decisions about protective behaviors (e.g., washing hands, wearing face masks, and distancing). However, mothers in all vaccine hesitancy categories were reluctant to accept the COVID-19 vaccine, mainly citing concerns about safety and efficacy and confusion over conflicting information as barriers to immediate vaccine acceptance. The findings indicate that mothers cannot be grouped based on their hesitancy over or acceptance of other vaccines for the purpose of assuming adherence to COVID-19 preventive behavior or early acceptance of the COVID-19 vaccine. In fact, our results show that mothers have higher scores on average (23.88 ± 5.65) than fathers (22.60 ± 4.26).

Furthermore, a specific increase in vaccination hesitancy is assumed in certain reference categories, such as level of education and employment. In fact, participants with a lower middle school education had higher average scores (24.51 ± 4.54) than those with a secondary school education (23.78 ± 6.51) or a university degree (22.52 ± 4.10). Participants who were housewives (24.20 ± 4.52) or unemployed (24 ± 4.08) showed higher averages than those who worked (23.96 ± 3.91), were employed by a public or private company (23.30 ± 4.66), were office workers (22.90 ± 7.90), or were freelancers (22.58 ± 4.53). These results are in line with findings in the literature [19,27] that consider the variables of schooling and work occupation as factors influencing vaccine hesitancy.

In Italy, the “RIV” and “IoVaccino” associations have published guidelines [28] that provide a series of specific reasons why parents are hesitant to vaccinate their children, and according to these guidelines, knowing the causes can lead to more effective vaccine support campaigns. First of all, the guidelines refer to previous personal experience, as people who have experienced an event over which they had no control, such as a natural disaster, losing a job, or contracting an illness, may be more likely to believe conspiracy theories [29]. Others may not rely on conspiracy theories but believe they can protect their families more effectively by adopting simpler prevention measures, such as an adequate diet, homeopathic remedies, or immune system stimulants. These parents need to understand how vaccines are manufactured and tested [30] and that they are the safest and most effective way to prevent disease. Even negative health experiences in the past (traumatic birth, disease misdiagnosis, etc.) can lead to distrust of doctors in general, so medical staff should be trained to use effective communication strategies and relationships to build trust with patients [31].

For some parents, the refusal to vaccinate is an extension of the belief that recommendations and requests to vaccinate are a sign of excessive intrusiveness on their autonomy by the government or are part of a profit-making plot hatched by pharmaceutical companies. Those who make decisions regarding vaccination schedules, according to these parents, would be hidden behind faceless commissions, organizations, and companies. Parents who believe in these things may need to learn more about both the decision-making process [32] and the historical process that led to the development of vaccination calendars [33].

Sometimes parents fear that if they get vaccinated and something goes wrong, they will be overwhelmed with guilt. These parents do not fully understand that even the choice not to vaccinate carries a risk [34] and that the risk of not vaccinating is very high, greater than the risk of having a severe reaction from a vaccine [35]. These parents may need to see how diseases affect children and understand how rare adverse reactions are, and how, by not vaccinating their children, they put them at greater risk for preventable diseases.

Moreover, the usefulness of investigating these factors to improve prevention strategies is also important for other population groups. For example, one study conducted in a sample of older people [20] showed that participants showed a good level of knowledge about COVID-19 characteristics and prevention but complained of a deterioration in their lifestyle-related habits as a result of the pandemic. It is necessary, therefore, to support this type of investigation in order to give more information to health prevention agencies.

### Limitations

The present investigation has methodological limitations. First, this investigation did not use a study design that could limit the possibility of making inferences about the temporality and causality of the observed relationships between variables. Second, it is a convenience-sampling-based study of a single center dedicated to COVID-19 vaccination and, therefore, it is possible that the results obtained may not be generalizable to the national situation but comparable to the few studies in the literature. Third, the data were collected with a self-reported questionnaire and, therefore, may be subject to recall or social desirability bias, as participants may have consciously selected positively oriented responses, thereby overestimating adherence to preventive measures of SARS-CoV-2 infection. However, participants were assured anonymity, and the results are likely to be authentic. Despite the limitations, this survey makes an important contribution to the topic, and the information will be useful for planning and implementing health education strategies.

## 5. Conclusions

The aspects related to the vaccination hesitancy of parents with regard to anti-COVID-19 vaccines detected by our survey are related to the poor ability to correctly perceive the risk of disease, the low quality of information on vaccines and the disease, and low levels of education. These results, in line with those found in vaccine hesitancy studies on other types of vaccines, need to be confirmed in larger populations and in different geographic areas. They should also prompt institutions and stakeholders to adopt communication tools aimed at improving trust in health institutions. The Eurobarometer Report considers it necessary to invest in order to “improve communication on this issue, especially on the advantages of vaccinations and their safety and efficacy”.

It is clear that it is necessary to provide the appropriate information that can explain the social value of vaccinations or correct misperceptions (for example, about the risks) and myths. This could be done through campaigns to change the perception of risk, campaigns that appeal to social motivations, campaigns that debunk the myths, and campaigns to promote vaccine acceptance [36,37,38,39].

Further studies are needed to define different communication strategies and to outline specific paths of knowledge for parents.

In conclusion, our results, in accordance with the evidence in the scientific literature, suggest the need to implement specific strategies to overcome parents’ vaccination hesitancy.

It is essential to identify specific interventions to support self-control or reduce external barriers that prevent vaccination. Vaccination strategies, which are always in progress (including new vaccines, pandemics, management of conflicts of interest, evaluation of risks and benefits, etc.), require pediatricians, who represent a fundamental point of reference for families, to remain constantly trained on this issue.

## Figures and Tables

**Table 1 vaccines-10-00958-t001:** Sociodemographic and parental characteristics and COVID-19 experience of total sample.

Main Categories	Variables		%
Sociodemographic	Gender	Female	64.5
		Male	35.4
	Marital status	Single	7.9
		Married	75.3
		Separated	8.4
		Divorced	4.8
		Cohabiting	3.6
	Level of schooling	Lower secondary school	17.3
		Secondary school	45.6
		University degree	32.1
		Other qualifications	5
	Employment	Public or private company	21
		Housewives	20.8
		Office workers	16.3
		Freelance professionals	15.8
		Other types of jobs	12
		Unemployed	7.9
		Workers	6.2
	Parent	Mother	64.5
		Father	31.4
		Other	4.1
Health status and COVID-19	Vaccination dose	First	2.9
		Second	15
		Booster	82.1
	SARS-CoV-2 infection	No	86.4
		Yes	13.6
		Asymptomatic	7.7
		Symptomatic	5.9

**Table 2 vaccines-10-00958-t002:** Frequency of answers to items on COVID-19 experience.

	How Concerned Are You Personally about the Problems Related to the COVID-19 Pandemic at the Moment?	How Likely Do You Think It Is That You Will Be Directly and Personally Affected by the COVID-19 Pandemic in the Next 6 Months?	How Likely Is It That Your Friends and Family in the Country You Currently Live in Will Be Directly Affected by the COVID-19 Pandemic in the Next 6 Months?
Not Worried at All	9.9%	8.2%	6.4%
A Little Worried	12.1%	17.7%	16.3%
Neither Very nor a Little Worried	17.7%	20.2%	20.1%
Quite Worried	36.8%	38.3%	40.1%
Very Worried	12%	8.3%	10.4%
Definitely Worried	7.4%	4.5%	4.5%
Extremely Worried	4.1%	2.8%	2.2%
Please indicate your degree of agreement or disagreement with the following statements:			
	“I will probably get sick with COVID-19”.	“COVID-19 sickness can be serious”.	“COVID-19 will not affect many people in Italy”.
Extremely Disagree	16.9%	5.6%	37.1%
Strongly Disagree	10.8%	7.1%	22.2%
Neither Agree nor Disagree	51.9%	29.9%	25.5%
Very Much in Agreement	16%	39.2%	9.2%
Extremely Agree	4.4%	18.2%	6%
What behaviors do you implement to prevent contagion?			
	Wash hands often/use sanitizing solutions	Maintain distance from other people	Always use a mask
Yes	94.9%	93.7%	96.5%
No	5.1%	6.4%	3.6%

**Table 3 vaccines-10-00958-t003:** Mean and standard deviation of PACV scores between specific categories/variables.

Variable		Mean ± SD
Gender	Female	23.88 ± 5.62
	Male	22.73 ± 4.39
Level of schooling	Lower secondary school	24.51 ± 4.54
	Secondary school	23.78 ± 6.51
	University degree	22.52 ± 4.10
	Other qualifications	23.20 ± 4.10
Employment	Employed by public or private company	23.30 ± 4.66
	Housewives	24.20 ± 4.52
	Office workers	22.90 ± 7.90
	Freelance professionals	22.58 ± 4.53
	Unemployed	24 ± 4.08
	Workers	23.96 ± 3.91
Parent	Mothers	23.88 ± 5.65
	Fathers	22.60 ± 4.26

## Data Availability

Written informed consent was obtained from the subjects in order to publish this paper. The archived data is not public and can be requested by writing to the corresponding author.

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
