# Peer review of "Anti-SARS-CoV-2 Vaccination Campaign: Risk Perception, Emotional States, and Vaccine Hesitancy in a Sample of Adolescents’ Vaccinated Parents in Southern Italy"

_vaccines, 2022, doi:10.3390/vaccines10060958_

Round 1
Reviewer 1 Report
- Title. It is my understanding that all the parents subject of this study had all been exposed to COVID-19 vaccines. Perhaps it should be mentioned also in title?
- Abstract. I think it is important to mention in the abstract that all parents interviewed had been exposed to vaccines. Line 13. ... among COVID-19 vaccines exposed parents of adolescents ... or just vaccinated parents .... I think it is important to make it clear upfront that 100% of these parents had themselves been exposed to vaccines, most even received a booster dose. Lines 20 and 22 - not very clear what these percentages in brackets actually mean (35.3% and 23.8%, respectively)
- Introduction. Would be here good to indicate in this section how much research on vaccines hesitancy in persons who themselves have been vaccinated has been carried out previously? I think it is an important difference if persons have been vaccinated/fully vaccinated/with boosters or if they are yet vaccine naive.
- Methods. Line 120, and Table1. 13.6 % had contracted the virus, 7.7% in asymptomatic form and 7.6% symptomatic. It might be my fault but 7.7 plus 7.6 is more than 13.6?
- Results. Layout of table 2 at present form is misleading, on the second page the headings of columns should be repeated, also info about the rows and numbers are not attached properly (likely this will be better when a final form is created?). Mean data in Table 2 show very little difference and SD is usually 20% or higher of mean. Is it indeed justified use terms "higher" in case of so small differences? They all also show that all data in Table 2 are homogeniously belonging to "low level hesitancy" (0-29) upper end.
- Discussion. The background of the study population of being 100% exposed to vaccines themselves should be discussed. Why the population who is vaccinated themselves has relatively high vaccine hesitancy about vaccination of their adolescent kids who are already pretty close to adult age? This study cannot be compared to studies with non-vaccinated populations.
- Conclusion. Not very much in line with the study and its results, I am not inclined to agree that the results of the study help in implementing specific strategies to overcome parents vaccination hesitancy in general. The study is in many ways far too COVID-19 specific and it also involved only parents with adolescent children. Some of the conclusions part seems also to be more relevant in discussions part.
Author Response
Authors thank the reviewer for important suggestions offered to improve our paper. All changes are in yellow.
- Title. It is my understanding that all the parents subject of this study had all been exposed to COVID-19 vaccines. Perhaps it should be mentioned also in title?
R: We have inserted the "vaccinated" specification referring to parents.
- Abstract. I think it is important to mention in the abstract that all parents interviewed had been exposed to vaccines. Line 13. ... among COVID-19 vaccines exposed parents of adolescents ... or just vaccinated parents .... I think it is important to make it clear upfront that 100% of these parents had themselves been exposed to vaccines, most even received a booster dose. Lines 20 and 22 - not very clear what these percentages in brackets actually mean (35.3% and 23.8%, respectively)
R: We have specified the percentages online 20-22 and made it clear that all parents have had the vaccination.
- Introduction. Would be here good to indicate in this section how much research on vaccines hesitancy in persons who themselves have been vaccinated has been carried out previously? I think it is an important difference if persons have been vaccinated/fully vaccinated/with boosters or if they are yet vaccine naive.
R: We have included in the introduction some studies related to data on vaccination hesitation in parents that were carried out before the approval of the vaccine in pediatric age.
- Methods. Line 120, and Table1. 13.6 % had contracted the virus, 7.7% in asymptomatic form and 7.6% symptomatic. It might be my fault but 7.7 plus 7.6 is more than 13.6?
R: We performed cross-tab analysis again and corrected the typing error in 5.9% for the symptomatic.
- Results. Layout of table 2 at present form is misleading, on the second page the headings of columns should be repeated, also info about the rows and numbers are not attached properly (likely this will be better when a final form is created?). Mean data in Table 2 show very little difference and SD is usually 20% or higher of mean. Is it indeed justified use terms "higher" in case of so small differences? They all also show that all data in Table 2 are homogeniously belonging to "low level hesitancy" (0-29) upper end.
R: We have redone table 2 and we hope that in this new form the reading of the data will be clearer. Furthermore, the understanding of the mean and SD differences is in Table 3. We have deliberately inserted these data divided into categories of variables to highlight the differences that emerged. We hope this detail is appreciated.
- Discussion. The background of the study population of being 100% exposed to vaccines themselves should be discussed. Why the population who is vaccinated themselves has relatively high vaccine hesitancy about vaccination of their adolescent kids who are already pretty close to adult age? This study cannot be compared to studies with non-vaccinated populations.
R: We have included some details that can be understood as a useful background to our data, regarding the pediatric vaccination campaign and the related hesitation.
- Conclusion. Not very much in line with the study and its results, I am not inclined to agree that the results of the study help in implementing specific strategies to overcome parents vaccination hesitancy in general. The study is in many ways far too COVID-19 specific and it also involved only parents with adolescent children. Some of the conclusions part seems also to be more relevant in discussions part.
R: We have revised the conclusions in the light of the new changes made and as suggested we have reshaped some parts in the "Discussions" paragraph.

Reviewer 2 Report
There are several missing gaps in this paper -
Not clear if both parents could reply to the survey
Critical to know when study done
ie were adolescent COVID vaccines available in the country when study done; if so how were they being offered ,
what were public health communications about these vaccines at the time
what were senior leaders in country saying about imm in this age group;
what was COVID doing at the time - cases in community, in this age group , what VOC ie effectiveneness of COVID vaccine in preventing infection vs preventing hospitalization and death.....
also not clear what COVID vaccine available - mRNA have increased risk myocarditis in this age group if male - was this known or not known at that time and was the vaccine mRNA?.....etc
Without this information the data are not interpretable ie context +++ matters
Curious use of "only" in line 203 - 85% is very good re "only 85% of
citizens believe that vaccines are effective in preventing infectious diseases " but need to know the time as not so effective with some VOC and does vary by type of COVID vaccine .....why timing so critical - discussion needs to be rewritten with this in mind
Not clear what is new here - discussion reads as not much - not sure same old same old merits publication BUT if can given time context - might be new findings compared to earlier papers in this area ie time and VOC and amount COVID around and infecting this age group etc
Author Response
The Authors would like to thank for the valuable methodological and content suggestions that we hope have improved our paper. All changes are in green.
There are several missing gaps in this paper - Not clear if both parents could reply to the survey
R: No, No, only one parents participated in the interview for each vaccination. We have specified this detail in the methodology section.
Critical to know when study done ie were adolescent COVID vaccines available in the country when study done; if so how were they being offered , what were public health communications about these vaccines at the time what were senior leaders in country saying about imm in this age group; what was COVID doing at the time - cases in community, in this age group , what VOC ie effectiveneness of COVID vaccine in preventing infection vs preventing hospitalization and death..… also not clear what COVID vaccine available - mRNA have increased risk myocarditis in this age group if male - was this known or not known at that time and was the vaccine mRNA?.....etc. Without this information the data are not interpretable ie context +++ matters.
R: In Introduction we have included the specification of the COVID-19 contagion and mortality data in pediatric age. In addition, we have included the details of when the administration of the COVID-19 vaccine in pediatric age was approved. In the "Discussions" paragraph, on the other hand, we have included further information relating to the vaccination campaign carried out for the pediatric age.
Curious use of "only" in line 203 - 85% is very good re "only 85% of citizens believe that vaccines are effective in preventing infectious diseases " but need to know the time as not so effective with some VOC and does vary by type of COVID vaccine .....why timing so critical - discussion needs to be rewritten with this in mind
R: We have included a specification of scientific literature relating to this data.
Not clear what is new here - discussion reads as not much - not sure same old same old merits publication BUT if can given time context - might be new findings compared to earlier papers in this area ie time and VOC and amount COVID around and infecting this age group etc
R: As also suggested by Reviewer 1, we have reworked both the "Discussions" and "Conclusions" paragraph. In this case the changes are in yellow.
